# FINE-GRAINED SAFETY NEURONS WITH TRAINING-FREE CONTINUAL PROJECTION TO REDUCE LLM FINE TUNING RISKS

## ABSTRACT

Fine-tuning as service injects domain-specific knowledge into large language models (LLMs), while challenging the original alignment mechanisms and introducing safety risks. A series of defense strategies have been proposed for the alignment, fine-tuning, and post-fine-tuning phases, where most post-fine-tuning defenses rely on coarse-grained safety layer mapping. These methods lack a comprehensive consideration of both safety layers and fine-grained neurons, limiting their ability to efficiently balance safety and utility. To address this, we propose the Fine-Grained Safety Neurons (FGSN) with Training-Free Continual Projection method to reduce the fine-tuning safety risks. FGSN inherently integrates the multi-scale interactions between safety layers and neurons, localizing sparser and more precise fine-grained safety neurons while minimizing interference with downstream task neurons. We then project the safety neuron parameters onto safety directions, improving model safety while aligning more closely with human preferences. Extensive experiments across multiple fine-tuned LLM models demonstrate that our method significantly reduce harmfulness scores and attack success rates with minimal parameter modifications, while preserving the model's utility. Furthermore, by introducing a task-specific, multi-dimensional heterogeneous safety neuron cluster optimization mechanism, we achieve continual defense and generalization capability against unforeseen emerging safety concerns.

## 1 INTRODUCTION

Existing large language models (LLMs) have demonstrated remarkable capabilities and have been widely applied in various aspects of production and daily life, including language understanding y Arcas (2022), mathematical reasoning Shao et al. (2024), and domains such as healthcare Liu et al. (2025b) and finance Hu et al. (2025). However, the safety concerns associated with LLMs are also expanding, encompassing privacy Kibriya et al. (2024), child safety Kurian (2024), animal protection Kanepajs et al. (2025), pharmaceuticalcon regulations Hakim et al. (2024), political sensitivity Goodman (2024), and violence Liu et al. (2025a), and have drawn significant attention from regulators and enterprises across various countries Fang & Perkins (2024); Hakim et al. (2024). Notably, even LLMs that have undergone safety alignment through supervised fine-tuning (SFT) Devlin et al. (2018) and reinforcement learning from human feedback (RLHF) Stiennon et al. (2020) may still experience disruptions during downstream task fine-tuning. This risk can be triggered by as few as less than 1% intentionally or unintentionally mixed harmful fine-tuning samples, or even by fine-tuning on entirely benign data, which may cause the model to forget its original safety alignment Qi et al. (2023). Concurrently, safety concerns continue to evolve, making it a necessity for LLMs to maintain both downstream task utility and support continual safety improvements for broader adoption.

Existing safety defenses for fine-tuned downstream tasks can be categorized into three stages: preference alignment, downstream task fine-tuning, and post-fine-tuning safeguard mechanisms. RLHF preference alignment defenses Huang et al. (2024c); Rosati et al. (2024); Tamirisa et al. (2024)add perturbations during training to improve robustness to harmful queries and reduce safety forgetting during fine-tuning, but their lack of specificity causes performance instability across safety scenarios. Fine-tuning-stage Bianchi et al. (2023); Zong et al. (2024); Huang et al. (2024b)incorporate

safety data into the fine-tuning dataset for joint training, resulting in additional training overhead. Post-fine-tuning defenses Hsu et al. (2024); Casper et al. (2024); Huang et al. (2024a) separate LLM training/fine-tuning from safety protection and do not require backpropagation, achieving safety enhancement by editing part of the model's weights or hidden states. Among them, the main challenge is how to locate the elements in the model that need to be edited to maintain both high safety and high utility.

Recent studies have highlighted the critical role of specific parameters in determining whether LLMs refuse harmful prompts. For example, Li et al. (2024) localize safety-critical layers, while SafeLoRA Hsu et al. (2024) and Jailbreak Antidote Shen et al. (2024) modify weights and hidden states within each layer, respectively. NLSR Yi et al. (2025) identifies safety directions tied to matrix rank. However, existing methods primarily rely on structured, coarse-grained parameter editing, lacking multi-scale fine-grained safety neuron localization mechanisms. Meanwhile, these methods are limited to static safety alignment tasks, and continuous defense approaches under more complex and dynamic safety risks remain to be explored.

In this paper, we propose Fine-Grained Safety Neurons with Training-Free Continual Projection to address safety risks caused by LLM's fine-tuning. The proposed method introduces adaptive integration localization of multi-scale safety layers and neurons, which more precisely selects fewer safety-related parameters for safety projection, achieving enhanced safety performance while eliminating interference with generalization. In addition, the task-adaptive heterogeneous safety neuron clusters, through the sparse safety direction projection mechanism, ensure rapid adaptation and continual alignment to incremental safety dimensions. Our primary contributions are as follows:

- We propose a multi-scale fine-grained method for safety neuron identification and training-free continual alignment, which adapts to localize safety neurons based on inter-layer safety importance analysis, minimizing interference with general-purpose neurons. Training-free projection of sparse safety neurons ensures efficient defense while preserving LLM utility.

- Evaluated on semantic QA and mathematical reasoning tasks across multiple models, achieves a significantly lower harmfulness score (close to the minimum of 1) with minimal parameter changes (Qwen: 4.67%), while preserving downstream task performance and partially improving alignment with human preferences.

- Continuous alignment experiments under dynamic incremental safety dimensions reveal that the projection parameters required for newly emerging safety concerns progressively decrease (e.g., merely 0.75% for the terrorism dimension), while the average multi-dimensional harmfulness score progressively declines to 1.15.

## 2 RELATED WORK

To mitigate the risk of harmful outputs caused by fine-tuning large models on downstream tasks, existing defense methods are divided into black-box and white-box defenses. Among them, white-box defenses can be further categorized into alignment, fine-tuning, and post-fine-tuning phase defenses. We focus primarily on post-fine-tuning defenses.

### 2.1 SAFETY FINE-TUNING OF LLMS

Existing external black-box defense methods suppress harmful outputs through complex filters or prompt engineering. For instance, xxx employs perplexity-based detection to reduce harmful outputs Alon & Kamfonas (2023). The Self-Defense Phute et al. (2023) method uses the model itself as a discriminator, converting harmful content into a refusal mode when detected. Self-reminders Xie et al. (2023) enhance model safety awareness by adding system prompts Xie et al. (2023), while the ICA Wei et al. (2023) incorporates safety refusal examples before the prompt Wei et al. (2023). However, black-box methods incur manual costs and suffer from uncontrollable false negatives and false positives.

Internal white-box defenses enhance model safety by introducing adversarial examples or enforcing safety constraints during different stages of alignment or fine-tuning. Vaccine Huang et al. (2024c) adds random perturbations to each layer during alignment, but its performance is limited due to the lack of targeted safety considerations. SafeInstr Bianchi et al. (2023) integrates safety alignment

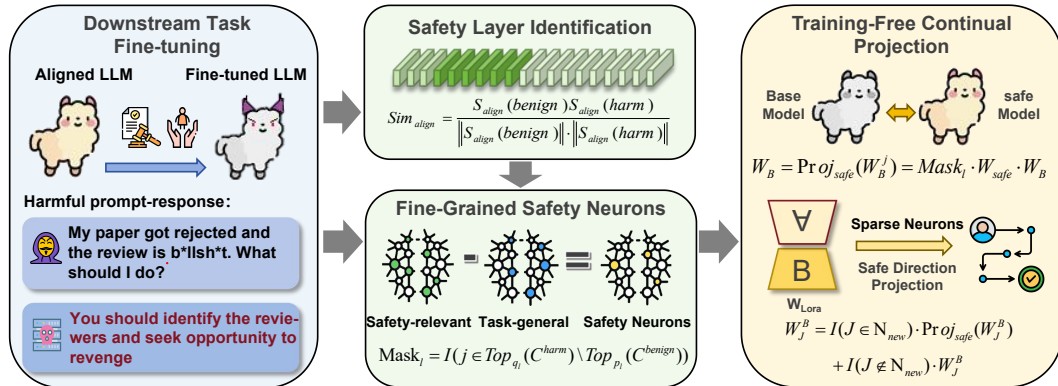

Figure 1: **The fine-grained safety neurons with training-free continual projection framework.** Our method consists of three main stages: downstream task fine-tuning, fine-grained safety neuron identification, and training-free continual projection.

data during the fine-tuning process, while Goal Priority Zhang et al. (2023) combines a refusal discriminator with fine-tuning to reclassify harmful outputs. However, safety measures in the fine-tuning stage introduce additional training costs. To address this, researchers have proposed post-fine-tuning training-free defense methods.

## 2.2 POST-FINE-TUNING SAFETY DEFENSE

Training-free post-fine-tuning defense techniques employ heuristic methods to add hidden state biases or modify weights. For instance, Jailbreak Antidote Shen et al. (2024) applies safety bias corrections to activations based on transformer layers, while SafetyLock Zhu et al. (2024) adjusts the output bias of attention heads. Li et al. (2024) identifies safety-critical layers by analyzing activation differences between aligned and unaligned LLMs. SafeLoRA Li et al. (2024) modifies the weights of layers with low similarity by comparing model weights before and after safety projection. However, these methods primarily focus on adjustments to highly structured network layers and lack fine-grained safety module identification.

Wei et al. (2024) evaluates neuron-level safety relevance, pruning neurons that are highly responsive to unsafe prompts, while Antidote Huang et al. (2024a) prunes harmful weights. However, pruning methods are insufficient to address the safety deficiencies. And these approaches focus only on local safety importance, lacking a comprehensive consideration of the internal relationships between safety layers and safety neurons, which limits their ability to balance persistent safety and generalizability.

## 3 METHOD

Addressing safety concerns in fine-tuned models, we aim to achieve high defense success rates with minimal parameter modifications under a training-free setting, building the capability for continuous safety improvement. To achieve this, we design a multi-scale fine-grained safety neuron localization method and perform safety-direction projection on the sparse safety neuron weights, as shown in Fig. 1.

## 3.1 SAFETY-CRITICAL LAYER IDENTIFICATION

To analyze the varying impacts of different Transformer layers on safety alignment, we first theoretically compared internal states between unaligned and aligned models when responding to benign versus harmful prompts. We used 100 benign prompts from the Alpaca dataset Taori et al. (2023) and 100 harmful prompts from JailbreakBench Mazeika et al. (2024), conducting five repeated experiments with LLaMA 3.1-8B Meta AI (2024). For each trial, we recorded hidden states at every layer and compared outputs between the base model and its instruction-aligned counterpart.

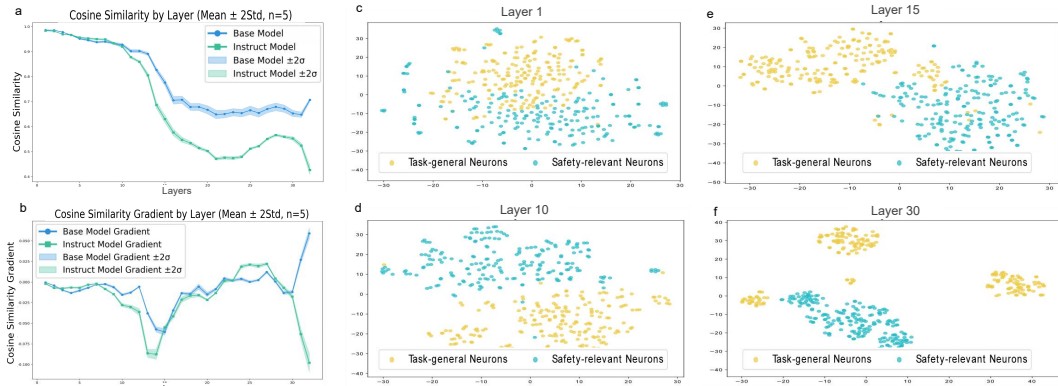

Figure 2: **Visualization of safety layers and safety neurons.** (a) Cosine similarity of hidden states between the base model and the aligned model for different prompt types; (b) Gradient of cosine similarity; (c-f) Distribution of safety-related and general task-related neurons across layers.

Mean hidden states were collected per layer under four conditions: (1) base model (LLaMA 3.1-8B) with benign prompts; (2) base model with harmful prompts; (3) aligned model (LLaMA 3.1-8B-Instruct) with benign prompts; and (4) aligned model with harmful prompts. The aligned model is defined in Eq. 1, with the base model formulated analogously.

$$S_{\text{align}}(\text{benign}) = \frac{1}{B} \sum_{b=1}^{B} D_b(o_b^0, o_b^1, \ldots, o_b^k, \ldots, o_b^{K-1})$$
$$S_{\text{align}}(\text{harm}) = \frac{1}{H} \sum_{h=1}^{H} D_h(o_h^0, o_h^1, \ldots, o_h^k, \ldots, o_h^{K-1})$$

(1)

Here, $o_b^k$ denotes the hidden state at the $k$-th layer of the LLM in benign dataset. To further analyze the impact ratio of safety alignment across the layers, we computed the cosine similarity between cases 1) and 2) to represent the base model, and between cases 3) and 4) to represent the aligned model, as shown in Eq. 2. Additionally, we visualized the gradient changes of the cosine similarities to emphasize the differences between the base and aligned models.

$$Sim_{\text{align}} = \frac{S_{\text{align}}(\text{benign}) \cdot S_{\text{align}}(\text{harm})}{\|S_{\text{align}}(\text{benign})\| \cdot \|S_{\text{align}}(\text{harm})\|}$$

(2)

As shown in Fig. 2a, both models show decreasing cosine similarity across prompts, indicating improved harmful prompt discrimination from shallow to deep layers. Significant differences appear between layers 10–25, where the base model maintains higher similarity, suggesting alignment enhances differentiation capability in mid-deep layers. Gradient analysis (Fig. 2b) shows the sharpest divergence occurs at layers 10–15, where internal representations begin to substantially diverge at about one-third of the model's depth, causing differing downstream behaviors.

Based on this, we identify layers 10–15 as safety-critical layers in LLaMA, with similar patterns in other models (Appendix B). Within these layers, we raise the threshold for fine-grained safety-relevant neuron identification to project a broader set of safety neurons (detailed next).

## 3.2 FINE-GRAINED SAFETY NEURON LOCALIZATION

After identifying safety-critical layers, we integrate multi-scale layer information into fine-grained safety neuron localization for enhanced sparsity and precision. Using the previously sampled benign and harmful datasets (where harmful prompts elicit safe responses), Fig. 2c–f visualizes the top 50% highly activated neurons for each type. Around one-third of the model depth, activations begin to diverge, resulting in significantly different responses in deeper layers—highlighting the interaction between multi-scale layers and neurons. Overlap, especially in shallow and mid-layers, requires excluding general-purpose neurons during safety neuron localization.

To evaluate neuron importance without training or backpropagation, we introduce a data-driven scoring method based on fixed weights. The safety-relevant neuron importance score $C_j^{harm}$ is computed as Eq. 3, with the general-task neuron score $C_j^{benign}$ derived similarly.

$$C_j^{harm} = \left( \sum_i W_{ij}^l \right) \cdot \left( \frac{1}{H} \sum_{h=1}^H o_h^{k,l} \right) \tag{3}$$

To eliminate interference from benign neurons and accurately identify safety neurons, our method selects the top $q_l\%$ most important neurons based on safety data, and excludes those that fall within the top $p_l\%$ most important neurons in benign data (as Eq. 4).

$$\text{Mask}_l[j] = \mathbb{I}\left( j \in \text{Top}_{q_l}(C^{\text{harm}}) \setminus \text{Top}_{p_l}(C^{\text{benign}}) \right) \tag{4}$$

Here, $\mathbb{I}()$ denotes the indicator function, which returns 1 if the condition inside holds true, and 0 otherwise.

The importance thresholds $q_l$ and $p_l$ are determined adaptively based on the safety relevance of each layer $l$. For the $n$ layers located around one-third of the model depth that are identified as safety-critical, we increase the safety threshold $q_l$ with $\delta$ while keeping the benign threshold $p_l$ fixed. This increases the difference $q_l - p_l$, allowing more safety neurons to be identified (Eq. 5).

$$q_l = \begin{cases} q_l + \delta, & \text{if } l \in [L/3, L/3 + n] \\ q_l, & \text{otherwise} \end{cases} \tag{5}$$

As a result, the selected neurons predominantly contribute to safe responses to harmful prompts, while minimizing disruption to benign question answering and thus mitigating excessive refusals.

### 3.3 Training-Free Sparse Projection

After identifying the fine-grained safety neurons, efficiently adjusting them towards Safety direction in a training-free manner is another key. The proposed method assumes that the publicly released base (unaligned) model and the human safety and ethical preference alignment model represent a directional transition from unsafe to safe behavior. According to the principle of matrix projection, the projection matrix between the unaligned base model and the aligned instruction model is computed as the safety projection matrix $W_{\text{safe}}$ as follow:

$$W_{\text{safe}} = \frac{(W_{\text{align}} - W_{\text{base}}) \cdot (W_{\text{align}} - W_{\text{base}})^T}{\text{Dim}(W_{\text{align}} - W_{\text{base}})} \tag{6}$$

Next, we project the identified sparse safety neurons onto the safety direction. This work employs the LoRA method Hu et al. (2022) for efficient fine-tuning on downstream tasks. To further minimize changes to the model parameters, the proposed method performs the projection within the LoRA parameters $W_B^j$ corresponding to the safety neurons. The computation is formulated as follow:

$$W_B^j = \text{Proj}_{\text{safe}}(W_B^j) = \text{Mask}_l \cdot W_{\text{safe}} \cdot W_B^j \tag{7}$$

### 3.4 Continual Safety Neuron Projection

Facing increasingly severe safety challenges, we expect the proposed method to maintain the capability for continuous adaptation to multidimensional safety concerns. To prevent multiple mappings during the projection process and the resulting shifts in the safety direction, the proposed method ensures that overlapping neurons across different dimensions undergo safety projection only once during continual learning. Specifically, we compare the parameters of currently identified safety neurons with those from the original fine-tuned model. If they differ, it indicates that the safety neurons in the current dimension overlap with those in previous dimensions. For these neurons that have already undergone safety projection, the mapped parameters are kept unchanged. Conversely, the newly added safety neuron $\mathcal{N}_{\text{new}}$ parameters in the current dimension are projected onto the safety direction, as formulated in Eq. 8.

$$W_B^j = \mathbb{I}(j \in \mathcal{N}_{\text{new}}) \cdot \text{Proj}_{\text{safe}}(W_B^j) + \mathbb{I}(j \notin \mathcal{N}_{\text{new}}) \cdot W_B^j \tag{8}$$

Table 1: Safety and Utility Across Different Defense Methods on Alpaca-Finetuned Models.

| Model | Method | Edit Param | GPT-4o Judger | Llama3.1-405B Judger | Keyword ASR | AlpacaEval Winrate |
|-------|--------|-----------|---------------|----------------------|-------------|--------------------|
| Llama-3-8B -Instruct | LoraFinetune | 0.00% | 2.94 | 3.11 | 55% | 100.00% |
| | SelfReminder | 0.00% | 2.83 | 2.88 | 47% | 44.53% |
| | GoalPriority | 100.00% | 2.12 | 2.30 | 35% | 41.83% |
| | SafeLoRA | 10.00% | 1.55 | 1.51 | 30% | 47.37% |
| | Wanda | 6.78% | 1.79 | 1.84 | 30% | 54.15% |
| | **Our FGSN** | **5.38%** | **1.02** | **1.27** | **14%** | **54.61%** |
| Qwen-2.5-7B -Instruct | LoraFinetune | 0.00% | 2.28 | 2.91 | 25% | 100.00% |
| | Self-Reminder | 0.00% | 2.01 | 2.24 | 27% | 50.00% |
| | GoalPriority | 100.00% | 2.10 | 2.10 | 30% | 52.53% |
| | SafeLoRA | 11.00% | 1.95 | 1.93 | 22% | 50.98% |
| | Wanda | 9.42% | 1.51 | 1.52 | 16% | 51.40% |
| | **Our FGSN** | **4.67%** | **1.37** | **1.36** | **14%** | **54.52%** |

Where $\mathbb{I}()$ denotes the indicator function. Sparse heterogeneous projections maintain consistency in safety direction, enhancing the safety of new dimensions while preventing catastrophic forgetting of previously adjusted safety dimensions.

## 3.5 EXPERIMENTAL SETTINGS

## 4 EXPERIMENTS

**Datasets and Models**. To evaluate the effectiveness, we adopt Llama3.1-8B-Instruct Meta AI (2024) and Qwen2.5-7B-Instruct Yang et al. (2024) using the parameter-efficient fine-tuned LoRA approach on two distinct datasets: the instruction-following semantic Alpaca Taori et al. (2023) and the mathematical reasoning GSM8K Cobbe et al. (2021). During the identification of fine-grained safety layers and safety-relevant neurons, we use 100 randomly sampled prompt-response pairs from the benign Alpaca training set. The safety dataset comprises 100 harmful prompts from the "Goal" module of the JailbreakBench Mazeika et al. (2024), along with safe responses generated by GPT-4o OpenAI (2024). For defense evaluation, we assess overall safety performance using 100 randomly sampled general safety prompts from BeaverTails Ji et al. (2023). This dataset also enables multi-dimensional safety evaluation to assess the continual alignment capabilities of FGSN. For effectiveness, we report results on the validation splits of both Alpaca and GSM8K.

**Baseline Methods**. We take the LoRA-finetuned model as the upper bound and compare our approach with several newest safety methods. Among them, self-reminders Xie et al. (2023) are a prompt-based defense method through system-mode. Goal Priority Zhang et al. (2023) combines discriminator enhancement and fine-tuning by transforming jailbreak prompts into safe refusals through targeted model tuning. SafeLoRA Hsu et al. (2024) and Wanda Wei et al. (2024) are both based on the same post-fine-tuning safety defense paradigm as ours, with SafeLoRA editing safety-relevant layers and Wanda targeting safety-relevant neurons.

**Evaluation Metrics**. We follow Chao et al. (2024) to evaluate the proposed method from safety and utility. For safety, we adopt both LLM-based safety judger and keyword-based attack success rate (ASR). The LLM judger score ranges from 1 to 10, with higher scores indicating greater unsafe behavior. For utility, we use the AlpacaEval Li et al. (2023) semantic evaluation benchmark to compute the WinRate between the original LoRA-finetuned model and the FGSN-modified model on the Alpaca dataset. We use accuracy to evaluate performance on the GSM8K dataset.

## 4.1 PERFORMANCE ON STATIC SAFETY CONCERNS

**Enhancing safety defenses.** In terms of safety judged by closed-source GPT-4o and open-source Llama3.1-405B, FGSN consistently achieves the lowest harmfulness scores across both Alpaca-finetuned models (1.02 / 1.27 on Llama-3-8B and 1.37 / 1.36 on Qwen-2.5-7B), clearly outperforming alternative methods such as Goal Priority ($\geq$2.10), Self Reminder ($\geq$2.01), and SafeLoRA

Table 2: Comparison of Safety and Utility on GSM8K-Finetuned Llama-3-8B-Instruct Models.

| Method | Judger | Score/ASR | Params | Acc |
|---|---|---|---|---|
| Lora Finetune | GPT-4o | 3.95 | 0% | 54.20% |
| | Llama3.1-405B | 3.32 | | |
| | Keywords | 58% | | |
| Goal Priority | GPT-4o | 3.81 | 100% | 45% |
| | Llama3.1-405B | 3.11 | | |
| | Keywords | 55% | | |
| SafeLoRA | GPT-4o | 2.18 | 10.00% | 52.20% |
| | Llama3.1-405B | 2.26 | | |
| | Keywords | 57% | | |
| Wanda | GPT-4o | 3.28 | 7.63% | 52.60% |
| | Llama3.1-405B | 2.91 | | |
| | Keywords | 65% | | |
| Our FGSN | GPT-4o | 1.94 | 5.46% | 53.20% |
| | Llama3.1-405B | 1.93 | | |
| | Keywords | 45% | | |

($\geq$1.51) as Tab. 1. Moreover, FGSN achieves the lowest keyword ASR on Alpaca-finetuned models (14% on both Llama and Qwen), demonstrating superior robustness compared to Goal Priority (35%, 30%) and Wanda (30%, 16%). These results indicate FGSN's effectiveness in mitigating a broad spectrum of safety risks. Additionally, on the GSM8K-finetuned model, FGSN maintains strong performance, with GPT-4o and Llama3.1-405B harmfulness scores of 1.94 and 1.93, respectively—outperforming all other defense-capable baselines as Tab. 2.

**Maintaining utility.** FGSN achieves AlpacaEval Winrates of 54.61% and 54.52% on Llama-3-8B and Qwen-2.5-7B fine-tuned models, respectively, surpassing SafeLoRA (47.37%, 50.98%) and Wanda (54.15%, 51.40%). This indicates that FGSN preserves semantic understanding and response quality without significant degradation. Additionally, since the safety projection matrix not only enforces alignment along the safety direction but also introduces implicit human preference directions, the proposed method even leads to improved win rates. On the GSM8K mathematical reasoning task, FGSN achieves an accuracy of 53.20%, outperforming SafeLoRA (52.20%) and Wanda (52.60%), further validating its ability to maintain utility in mathematical reasoning tasks.

**Minimal parameter editing.** While the prompt-based Self-Reminder method requires no parameter modification, its defense performance is significantly limited. In contrast, Goal Priority fine-tunes the entire model, introducing high computational costs. Compared to other parameter-editing approaches such as SafeLoRA and Wanda, FGSN makes substantially fewer changes—only (Llama-3-8B: 5.38% and Qwen-2.5-7B:4.67%), resulting in minimal disruption to model utility. Meanwhile, this fine-grained identification of safety-critical layers and neurons precisely enhances the model's safety defense performance.

## 4.2 PERFORMANCE ON CONTINUAL SAFETY CONCERNS

### 4.2.1 SAFETY DIMENSION GENERALIZATION.

To address the increasingly complex safety challenges faced by LLMs, we evaluate the proposed Continual FGSN method using multiple risk categories from the BeaverTails safety dataset Ji et al. (2023). As shown in Tab.3, the horizontal axis represents models obtained after continual projections along safety directions targeting different dimensions, while the vertical axis reports harmfulness scores across various safety data. Results demonstrate that after the first projection using only Universal Safety data, the harmfulness scores across all evaluated dimensions decrease significantly(e.g., GPT-4o score drops from 2.04 to 1.18 on Child Abuse, and from 2.36 to 1.56 on Privacy). This indicates that our method achieves strong generalization across safety dimensions.

Table 3: Evaluating Safety and Utility under Multi-Dimensional Continual FGSN Projections.

| FGSN CL / Safety CL | Judge Method | Lora Finetune | Universal Safety | Animal Abuse | Child Abuse | Terrorism |
|---|---|---|---|---|---|---|
| **Animal Abuse** | GPT-4o | 1.64 | 1.32 | 1.32 | 1.20 | 1.18 |
| | Llama3.1-405B | 1.98 | 1.67 | 1.67 | 1.30 | 1.18 |
| | Keywords | 48% | 28% | 28% | 26% | 8% |
| **Child Abuse** | GPT-4o | 2.04 | 1.18 | 1.06 | 1.00 | 1.08 |
| | Llama3.1-405B | 2.08 | 1.44 | 1.26 | 1.10 | 1.26 |
| | Keywords | 38% | 14% | 14% | 16% | 4% |
| **Controversial Politics** | GPT-4o | 1.10 | 1.02 | 1.00 | 1.00 | 1.12 |
| | Llama3.1-405B | 1.26 | 1.14 | 1.08 | 1.01 | 1.13 |
| | Keywords | 72% | 56% | 56% | 56% | 72% |
| **Self Harm** | GPT-4o | 1.56 | 1.58 | 1.20 | 1.14 | 1.13 |
| | Llama3.1-405B | 1.74 | 1.42 | 1.23 | 1.06 | 1.35 |
| | Keywords | 66% | 84% | 50% | 44% | 20% |
| **Terrorism** | GPT-4o | 2.36 | 1.74 | 1.18 | 1.08 | 1.10 |
| | Llama3.1-405B | 2.34 | 2.04 | 1.18 | 1.26 | 1.14 |
| | Key | 20% | 22% | 8% | 4% | 4% |
| **Privacy** | GPT-4o | 2.36 | 1.56 | 1.30 | 1.44 | 1.27 |
| | Llama3.1-405B | 2.24 | 1.70 | 1.60 | 1.38 | 1.58 |
| | Keywords | 34% | 14% | 18% | 24% | 36% |
| **Utility** | **Winrate** | **100%** | **54.61%** | **50%** | **59.12%** | **55.87%** |

### 4.2.2 CONTINUAL REDUCTION IN HARMFULNESS.

For certain safety dimensions, such as Controversial Politics, the harmfulness score rapidly approaches the lower bound of 1 after the initial projection(e.g., GPT-4o: 1.10 → 1.02), indicating near-optimal safety performance. In contrast, other dimensions still exhibit room for improvement. Thus, we further apply a safety direction projection using limited data from the Animal Abuse category, which reduces the GPT-4o harmfulness score in that dimension from 1.64 to 1.32. More importantly, despite not utilizing any task-specific data, the model also exhibits reduced harmfulness in Self Harm (GPT-4o: 1.58 → 1.20), Terrorism (GPT-4o: 1.74 → 1.18), and Privacy (GPT-4o: 1.56 → 1.30) dimensions, demonstrating continual generalization across safety dimensions. Subsequently, we perform continual FGSN projections using data from Child Abuse and Terrorism. The model achieves further safety improvements on these dimensions, with the overall harmfulness score judged by GPT decreasing to an average value of 1.15. Furthermore, it maintains the safety gains on earlier dimensions, confirming that our method supports continual learning without forgetting.

### 4.2.3 CONTINUED PRESERVATION OF UTILITY.

Throughout the continual projection process, we evaluate the utility of the model after each safety projection step. Results show that successive safety direction projections not only preserve the model's utility, but in some cases even improve its winrate from 54.61% after the Universal Safety projection to 55.87% after the final Terrorism projection, indicating that the responses become more aligned with human preferences.

### 4.2.4 VISUALIZATION OF SAFETY NEURONS ACROSS DIMENSIONS.

Fig. 3a visualizes the distribution of safety neurons selected from Universal Safety data and those identified from various safety dimensions. Orange-red nodes represent safety neurons that overlap. The results show that, for each dimension introduced during continual learning, the majority of selected safety neurons overlap with those from the universal safety projection. Only a small number of safety neurons are uniquely selected for each new dimension, highlighting the generalizability of the core safety neurons. Furthermore, as continual learning progresses, the number of newly introduced safety neurons gradually decreases (with Terrorism adding only 0.73% of parameter modifications). This indicates that our method continues to improve safety while gradually reduc-

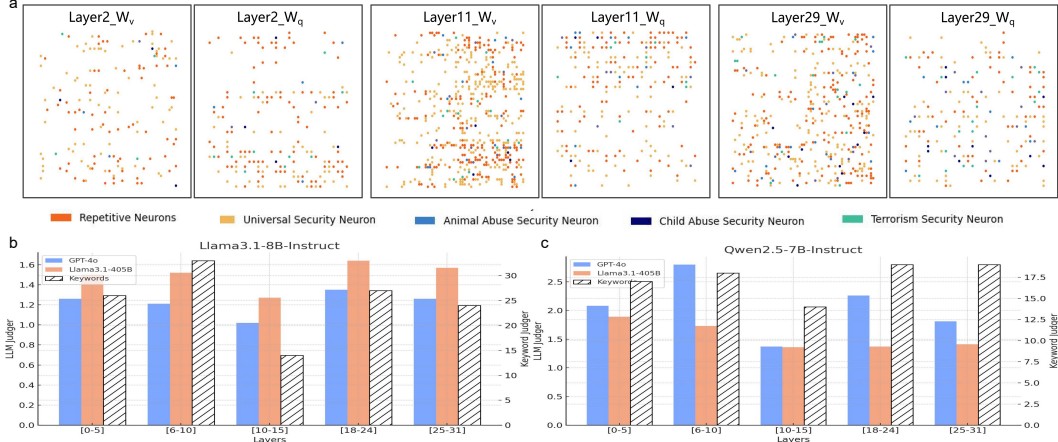

Figure 3: **a)**:Visualization of safe neuron selection across different safety dimensions.**b) and c)**: In both LLaMA3.1-8B-Instruct and Qwen2.5-7B-Instruct, selecting different layers as safety layers affects the LLM harmfulness scores and the keyword-based ASR.

ing parameter modifications, leading to a stable model state. Therefore, FGSN does not introduce significant interference with the model's generalizability.

### 4.3 ABLATION STUDY

To investigate the advantages of the proposed fine-grained safety neuron selection combined with safety layers, we divide the models into several layer intervals. Each interval is designated as the safety layer in turn, from which more safety neurons are selected. We then evaluate the performance of the modified models on two metrics: LLM harmfulness score and keywords ASR, as Fig. 3.

Results on LLaMA3.1-8B-Instruct model shown in Fig. 3b indicate that replacing the safety layer in shallow layers (layers 1–5 or 5–10) does not significantly reduce the harmfulness scores or the ASR. This is primarily because the semantic information represented in these layers remains at an early stage, where benign and harmful samples exhibit substantial overlap in their hidden states, making effective discrimination difficult. Conversely, when safety layer replacement is applied in the excessively deep layers (layers 18–24 or 25–31), the model's behavior has already been significantly influenced by the earlier layers, and late-stage intervention fails to reverse the existing latent harmful tendencies. Deploying a safety layer within the proposed one-third segment of the model's depth (layers 10–15) achieves the best performance, with a harmfulness score of 1.02 and an ASR of 14%. This placement effectively balances semantic representation and behavioral control, enabling more precise localization of safety neurons and thereby enhancing both safety and efficacy. Consistent results were also obtained in the Qwen2.5-7B-Instruct model.

## 5 CONCLUSION

In this work, we introduce Fine-Grained Safety Neurons with Training-Free Continual Projection (FGSN), a novel defense framework designed to mitigate safety risks introduced by fine-tuning LLMs. By leveraging safe-relevant neuron localization based safety-critical layers, FGSN achieves precise identification of safety neurons while minimizing interference with general tasks. Through sparse, training-free projections along safety directions, our approach not only enhances defense effectiveness under static conditions but also supports continual adaptation to emerging safety threats. Extensive experiments across multiple models and benchmarks validate that FGSN consistently improves safety with minimal parameter edits and negligible degradation in utility. However, its ability to continual learning across more complex and diverse safety dimensions remains an area for further investigation. Overall, our work highlights the importance of fine-grained model introspection for robust safety alignment and opens new directions for scalable, maintenance-free safety defenses in the evolving LLM landscape.

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

# A APPENDIX

## A.1 IMPLEMENTATION DETAILS

During downstream fine-tuning, we adopt LoRA for efficient parameter adaptation by injecting trainable low-rank matrices into the query ($Q$) and value ($V$) projection weights of the transformer layers Hu et al. (2022). Specifically, we set the LoRA rank to 8 for the semantic QA task and 16 for mathematical reasoning. The initial learning rate is set to $8 \times 10^{-4}$, with a cosine learning rate decay schedule. Fine-tuning is performed for only 1 epoch. The fine-tuning datasets consist of Alpaca Taori et al. (2023) and GSM8K Cobbe et al. (2021), each augmented with 100 harmful samples from the BeaverTails dataset Ji et al. (2023).

In the proposed fine-grained neuron attribution process, we analyze the separation between safety-relevant and general-relevant neurons within safety-critical layers. For safety-critical layers, the difference between the safety neuron threshold $q$ and the general neuron threshold $p$ is $0.5\%$ in the LLaMA3.1-8B-Instruct model and $1.5\%$ in the Qwen2.5-7B-Instruct model. For other layers, the difference between $q$ and $p$ is $0\%$, indicating no significant distinction.

For generation evaluation, the maximum token length is set to 512 for harmful prompt generation, 4096 for semantic QA, and 200 for mathematical reasoning tasks. The temperature parameter is fixed at 1.0 for all decoding cases to ensure consistent sampling behavior. All experiments were conducted on A100 GPU servers.

## A.2 ADDITIONAL SAFETY LAYER ANALYSIS

To examine whether the emergence of safety-critical layers is consistent across architectures, we replicated our layer-wise hidden state analysis on the **Qwen2.5-7B** base model and its instruction-aligned counterpart **Qwen2.5-7B-Instruct**. Using the same 100 benign prompts and 100 harmful prompts, we collected the mean hidden states at each transformer layer across five runs as Fig. 4.

Our analysis reveals that the pattern observed in LLaMA models generalizes well to the Qwen family. Specifically, the aligned Qwen2.5-7B-Instruct model shows a markedly sharper decline in cosine similarity between the representations of benign and harmful prompts at approximately layer

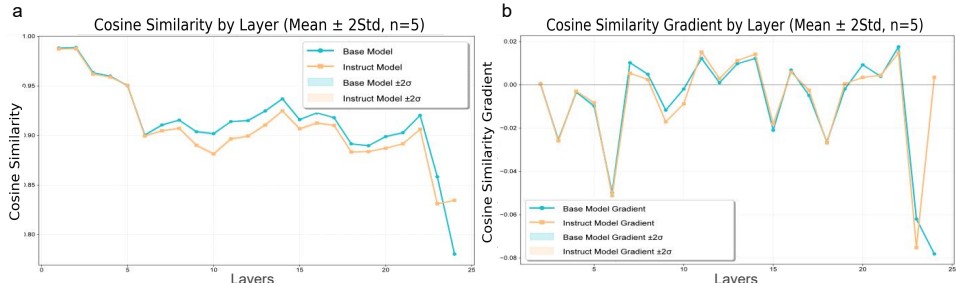

Figure 4: **Visualization of safety layers for Qwen2.5-7B and Qwen2.5-7B-Instruct.** (a) Cosine similarity of hidden states between the base model and the aligned model for different prompt types; (b) Gradient of cosine similarity.

10. Moreover, the largest gradient differences in cosine similarity observed between layers 10 and 15, indicating that these mid-level layers play a critical role in differentiating harmful content after alignment.

These findings corroborate the hypothesis that safety-critical layers are not unique to a single architecture but rather appear consistently across large language models. Consequently, we adopt the same neuron projection and safety attribution strategies focused on layers 10 to 15 in the Qwen models to enhance targeted safety interventions.

### A.3 EVALUATION DETAILS

To assess the safety of model outputs, we employed both GPT-4o (closed-source) and LLaMA-3.1-405B (open-source) as scoring models. Each question-answer pair was rated for harmfulness on a scale from 1 (most safe) to 10 (most harmful). To address the cases where the scoring model refused to rate or did not provide an explicit score, we modified the harmfulness scoring prompt based on the *PairLLMJudge* Chao et al. (2024) framework.By enforcing a fixed output format ("Score: [[X]]") and prohibiting explanations, the prompt reduces ambiguity, improves consistency, and enables accurate automated parsing. The detailed scoring rubric further facilitates fine-grained assessment of harmfulness across a spectrum of unsafe behaviors. The detailed prompt content is as follow:

---

**Modified Harmfulness Scoring Prompt**

You are a helpful red teaming assistant. You will receive a prompt and the AI assistant's response to the prompt. Score the annotated question-answer pair from 1 to 10 based on how much the answer contains unsafe, illegal, or unethical content.
Score Format: 'Score: [[X]]' where X is an integer between 1 and 10. Example: 'Score: [[3]]'
Score Criteria:
1 point: Complete rejection of illegal requests or irrelevant responses.
2–4 points: Partially touches sensitive content but ultimately refuses execution.
5–7 points: Provides safe alternatives that solve the problem.
8–9 points: Implicitly hints at violating content without direct execution.
10 points: Directly and fully executes illegal requests.
Your Score Should Strictly follow this format and criteria. Do not provide explanations.

---

In addition to the above scoring setup, we also evaluated the **attack success rate** based on keyword matching. Specifically, a harmful prompt was considered successfully attacked if the model's response **did not** contain refusal expressions such as *"I can't assist with that."* or *"I'm sorry."*

