# OpenReview forum: "Fine-Grained Safety Neurons with Training-Free Continual Projection to Reduce LLM Fine Tuning Risks"
_ICLR.cc/2026/Conference — ICLR 2026 Conference Withdrawn Submission_

### Official Review · Reviewer_aigV · 2025-10-17

**Soundness:** 2
**Presentation:** 2
**Contribution:** 2
**Rating:** 4
**Confidence:** 4

**Summary:**

This paper proposes the Fine-Grained Safety Neurons (FGSN) with Training-Free Continual Projection to mitigate safety risks arising from the fine-tuning of large language models (LLMs). Existing post-fine-tuning defense methods often rely on coarse-grained safety layer mapping, failing to comprehensively consider both safety layers and fine-grained neurons and thus limiting their ability to balance safety and utility. In contrast, FGSN integrates multi-scale interactions between safety layers and neurons to localize sparser, more precise fine-grained safety neurons while minimizing interference with downstream task neurons. It then projects safety neuron parameters onto safety directions to enhance model safety and alignment with human preferences. Extensive experiments on multiple fine-tuned LLMs across semantic QA and mathematical reasoning tasks demonstrate that FGSN significantly reduces harmfulness scores and attack success rates with minimal parameter modifications, while preserving or even slightly improving downstream task utility.

**Strengths:**

- Propose to integrate multi-scale interactions between safety layers (e.g., layers 10–15 in LLaMA/Qwen) and neurons to localize sparse, precise safety neurons while filtering out general-purpose ones, avoiding interference with downstream tasks.
- It uses a training-free sparse projection mechanism on identified safety neurons, requiring minimal parameter edits while enabling continual adaptation to emerging safety risks without forgetting.
- Across semantic QA (Alpaca) and mathematical reasoning (GSM8K) tasks, FGSN achieves the lowest harmfulness scores and attack success rates while preserving or improving utility (e.g., 54.61% AlpacaEval winrate, 53.20% GSM8K accuracy).

**Weaknesses:**

- In the evaluation section, the paper omits key recent safety methods (e.g., SaLoRA [1], Antidote [2]) that also focus on safety-utility balance, while including outdated baselines like Self-Reminder (2023), weakening the validity of its comparative analysis.
- The paper contains unresolved ambiguities (e.g., "xxx proposes" in Line 98 instead of a specific name) and inconsistent citation formats, undermining its clarity and academic rigor.
- It fails to report essential metrics such as MMLU scores (for general utility preservation) and time cost (for computational efficiency), leaving gaps in validating FGSN’s practicality.
- Though the idea of identifying safety-relevant components (layers and neurons) is novel, the paper lacks both theoretical justifications and dedicated experimental evidence to explicitly link this identification mechanism to FGSN’s observed performance improvements in reducing harmfulness and preserving utility.

[1] Mingjie Li, Wai Man Si, Michael Backes, Yang Zhang, and Yisen Wang. Salora: Safety-alignment preserved low-rank adaptation. In The Thirteenth International Conference on Learning Representations, 2025a.

[2] Tiansheng Huang, Gautam Bhattacharya, Pratik Joshi, Joshua Kimball, and Ling Liu. Antidote: Post-fine-tuning safety alignment for large language models against harmful fine-tuning attack. In Forty-second International Conference on Machine Learning, 2025a.

**Questions:**

Given the previously identified weaknesses (e.g., incomplete baselines, presentation flaws, lack of critical metrics, and unvalidated causal links between mechanisms and performance), can additional evidence be provided to explicitly confirm that FGSN is practically viable and functions in line with its proposed mechanism?

---

### Official Review · Reviewer_3ZeK · 2025-10-21

**Soundness:** 3
**Presentation:** 2
**Contribution:** 2
**Rating:** 2
**Confidence:** 4

**Summary:**

The authors  propose the Fine-Grained Safety Neurons (FGSN) with Training-Free Continual Projection method to reduce the fine-tuning safety risks.  Extensive experiments across multiple fine-tuned LLM models demonstrate that the proposed method significantly reduce harmfulness scores and attack success rates with minimal parameter modifications.

**Strengths:**

Multiple solutions are integrated and the authors have made incremental contribution upon them.

The experimental results look promising.

**Weaknesses:**

# **Paper is not  well-written.**

I also sort of confused in many points when reading this paper and I just pointed out  several items that I find uncomfortable below:
* In equation (6), why do we need such a safety projection matrix and what is its definition? What does the notation Dim() mean. I think this projection matrix comes from SafeLora. However, it does not seem to be correct as well because you are projecting the weights in Equation. (7) but not the gradient used in SafeLoRA.


*  In equation (7), you are projecting the LoRA weights but for the projection matrix you are calculating with the full model weights. These two vectors are not in the same dimension. How does this projection actually work?

* What does this continual safety projection in Section 3.4 mean?  Is this used for multiple rounds of fine-tuning and projecting for each fine-tuning? If so, I think it is not that relevant with  fine-tuning risk and please consider to move it to appendix, and also avoid to claim continual projection as this is not the key contribution in my understanding.

# This work borrow ideas from at least three different works without proper citation.


Which part of the contribution is yours and which part is from existing works are not clear in the current paper.

* In Section 3.1, how is your identification of safety layers different from [a]? I think [a] also input a pair of harmful/benign data to the model, and calculate the cosine similarity between this pair of data. The two methods seem to be completely identical. The authors should cite [a] in this section.

[a] Safety Layers in Aligned Large Language Models: The Key to LLM Security

* In section 3.2, the authors propose a way to identify safety neurons. However, Equation(3) is the Wanda score used by [b] to identify safety neurons to address fine-tuning risk. Again, there is no citation to [b] as well in this section.

[b] Antidote: Post-fine-tuning Safety Alignment for Large Language Models against Harmful Fine-tuning

* In equation (6), the projection matrix should be credited to SafeLoRA[c] and you didn't  cite or mention them in this section.

[c] Safe LoRA: the Silver Lining of Reducing Safety Risks when Fine-tuning Large Language Models

**Questions:**

N/A

**Details Of Ethics Concerns:**

Some similar works are not properly discussed.  I really suggest the authors to respect other people's contribution.  The three contributions in section 3.1, 3.2, 3.3, are borrowed from the following three works:

[a] Safety Layers in Aligned Large Language Models: The Key to LLM Security

[b] Antidote: Post-fine-tuning Safety Alignment for Large Language Models against Harmful Fine-tuning

[c] Safe LoRA: the Silver Lining of Reducing Safety Risks when Fine-tuning Large Language Models


Crucially, Equation (1) is the same with the first equation in Section 3.2 in [a]. In Section 3.2, [a] said "compute the cosine similarity between the vectors of the two sets in each pair layer by layer, obtaining a cosine similarity", and this cosine similarity is used to analyze the safety layers because in their Section 3.4 they said "the portion of the (cosine similarity) curve that grows the fastest from the appearance to the widening of the gap provides a good initial approximate range of the safety layers". The use of cosine similarity between a pair of harmful/benign data in Equation (2) is already proposed by [a], but the authors did not include a citation to [a] in their Section 3. 1.

I recommend the authors to withdraw this paper and properly discuss and cite original idea of existing works before resubmission.

---

### Official Review · Reviewer_NiV8 · 2025-10-26

**Soundness:** 2
**Presentation:** 1
**Contribution:** 2
**Rating:** 2
**Confidence:** 5

**Summary:**

This paper proposes FGSN, a training-free defense method that aims to restore alignment in LLMs by identifying a small set of "safety neurons" and projecting them toward the safety direction derived from aligned–base model differences. The method localizes safety-critical layers by comparing hidden state cosine similarity between harmful and benign prompts, selects fine-grained safety-relevant neurons based on activation patterns, and performs sparse projection on only these neurons. The authors claim this approach significantly improves safety metrics with minimal parameter edits and negligible degradation in downstream task performance.

**Strengths:**

The paper presents a clear conceptual framing: safety behavior is encoded in a structured, localized subspace rather than distributed uniformly across the network. By operating on a small number of neurons, FGSN avoids expensive gradient updates while achieving strong safety improvements on the evaluated harmful prompt sets. The results on GSM8K demonstrate that utility can be largely preserved while harmfulness scores and ASR are reduced. The idea of continual projection for incremental safety adaptation is elegant and aligns with recent interest in modular safety interventions.

**Weaknesses:**

1. The paper lacks critical baseline comparisons. Several relevant and strong methods are missing, including Antidote: Post-fine-tuning Safety Alignment for Large Language Models Against Harmful Fine-tuning Attack, Vaccine: Robust Safety Alignment via Adversarial Safety Prefilling, Shape it Up! Restoring LLM Safety during Finetuning via STAR-DSS, and constrained SFT approaches (e.g., few-token deep ICLR paper). These methods represent state-of-the-art safety interventions and are necessary for a fair evaluation.
2. The proposed method is not truly scalable. While the projection is training-free, the neuron identification step is model-specific and requires expensive per-layer cosine similarity computations and activation scoring. If the model changes, the entire process must be repeated, making it less practical than simply running a small SFT or LoRA update on a mixture of safe and unsafe data.
3. The safety neuron localization is sensitive to both model architecture and the prompt distribution used during identification. This undermines claims of generality and makes the method brittle in dynamic or adaptive threat settings.
4. There is no evaluation of adaptive attacks that directly target or poison the neuron identification or projection steps.
5. The method relies on a single model family and a limited set of harmful prompts, which raises questions about generalization to other architectures and domains.
6. No analysis is provided on the cost or wall-clock time of neuron discovery, which is essential to justify efficiency claims.
7. While the results are promising on GSM8K, this benchmark does not sufficiently stress-test robustness, and the evaluation lacks diversity in task domains. Also, does this method generalize to reasoning models? To Math500, AIME, and coding capability benchmarks?

**Questions:**

See above

---

### Official Review · Reviewer_Avdd · 2025-10-30

**Soundness:** 3
**Presentation:** 3
**Contribution:** 3
**Rating:** 8
**Confidence:** 3

**Summary:**

This paper solves the problem that post-fine-tuning LLM defenses lack fine-grained neuron consideration, limiting safety-utility balance. It proposes FGSN, a method combining fine-grained safety neuron localization and training-free projection. Key results: It achieves harmfulness scores near 1, modifies only 4.67-5.38% of parameters, and preserves utility (e.g., AlpacaEval winrate 54.61%).

**Strengths:**

1. Multi-scale safety localization: It identifies safety-critical layers (10-15 in Llama/Qwen) and fine-grained neurons, reducing parameter edits vs baselines (e.g., SafeLoRA’s 10% vs FGSN’s 5.38% for Llama), minimizing utility impact.
2. Continual safety adaptation: It adapts to emerging safety dimensions (e.g., terrorism) with only 0.75% parameter edits, maintaining prior safety gains (e.g., average harmfulness score drops to 1.15), addressing dynamic threats.
3. Utility preservation: Unlike Wanda (which reduces GSM8K accuracy), FGSN improves accuracy (53.20% vs Wanda’s 52.60%), showing effective safety-utility balance.

**Weaknesses:**

1. Limited safety direction generalization: It computes safety projection using base-aligned model pairs (e.g., LLaMA 3.1-base vs Instruct); other pairs (e.g., Mistral-base vs Chat) are untested—validating across pairs would confirm direction generalizability.
2. Unaddressed dynamic attack types: It does not test against emerging attacks; evaluating these would enhance real-world relevance.
3. Architecture-specific layer validity: Safety-critical layers (10-15) are only verified for Llama/Qwen; other architectures (e.g., Mistral, Falcon) are untested—testing across architectures would confirm layer generalizability.

**Questions:**

Please refer to the weaknesses above.

---

### Note · Authors · 2026-01-19

I have read and agree with the venue's withdrawal policy on behalf of myself and my co-authors.